# Theoretical and Experimental Research on the Mode Modulation Regulation for the Mode-Tunable Vortex Laser Based on Mode Conversion and Intra-Cavity Modulation

Shibing Lin [1,2] , Deen Wang [1,2,3], Shaonan Kang [1,2], Yamin Zheng [1,2] and Lei Huang [1,2,*]

1   Key Laboratory of Photonic Control Technology, Tsinghua University, Ministry of Education, Beijing 100084, China; linsb19@mails.tsinghua.edu.cn (S.L.); wde18@mails.tsinghua.edu.cn (D.W.); kangsn@mail.tsinghua.edu.cn (S.K.); zhengym20@mails.tsinghua.edu.cn (Y.Z.)
2   State Key Laboratory of Precision Measurement Technology and Instruments, Department of Precision Instrument, Tsinghua University, Beijing 100084, China
3   Research Center of Laser Fusion, China Academy of Engineering Physics, Mianyang 621900, China
*   Correspondence: hl@tsinghua.edu.cn

**Abstract:** The vortex laser beam has been widely applied in many fields for its unique properties. However, researchers have to conduct extensive and recurring experiments to find the modulation abilities of the vortex beam modes for a given resonant cavity. In this paper, a mode modulation regulation acquisition (MORA) method, investigating the relationship between the modes of the vortex beam and modulation parameters, is proposed and verified. A typical mode-tunable vortex laser, consisting of a classic plano-concave straight cavity, a vortex beam generation beamline, and a reference beam acquisition beamline, is used as the analysis and experiment object. The principle and working process of the MORA method is analyzed in the simulation, and its validity is verified in the experiment. Based on the obtained theoretical relationship between the modes of the vortex beam and modulation parameters, the MORA method could be used to help researchers in designing the practical vortex laser with target vortex beams output by optimizing the structure of the vortex laser, selecting the suitable intra-cavity modulation elements (IMEs), and pre-positioning the location of the IMEs.

**Keywords:** vortex laser beam; mode modulation regulation; intra-cavity modulation elements

## 1. Introduction

Vortex beams with a spiral phase distribution, carrying the orbital angular momentum (OAM), have been widely applied in many areas, including optical tweezers [1], particle manipulation [2], photo entanglement [3], and quantum communication [4]. Various methods have been reported to generate the OAM-tunable vortex beam, including ring pump [5,6], double cavity structure [7], metasurfaces [8], Bragg grating [9], and mode converters [10]. Among these methods, the mode conversion based on an astigmatic mode converter (AMC), which consists of a pair of cylindrical lenses and could convert the arbitrary Hermite–Gaussian (HG) beam to a corresponding Laguerre–Gaussian (LG) beam (i.e., a type of vortex beam), has high conversion efficiency and could achieve the high-order vortex beam. In particular, the type of $HG_{0,n}$ beam could be converted to the type of $LG_{0,n}$ vortex beam with $\pm n\hbar$ OAM [11].

It is well known that the HG mode is the intrinsic mode of a resonant cavity, and the high-order HG mode could be generated from the resonant cavity based on intra-cavity modulation. Many methods have been reported to generate the mode-tunable HG beam using the intra-cavity modulation elements (IMEs), including the moveable concave mirror [10], deformable mirror [12], moveable cylindrical lens [13], and spatial light modulation [14,15]. A moveable concave cavity mirror is used as an IME in a plano-concave cavity [10], and up to $HG_{0,15}$ beam could be generated by moving the dual off-axis

displacement of the concave mirror. Two moveable cylindrical lenses are inserted into a resonant cavity as IMEs, and complex HG modes (i.e., $HG_{m,n}$ with index $m \neq 0$ and $n \neq 0$) could be obtained by moving the cylindrical lenses in certain directions [13]. For these two methods, the high-order HG beam could be generated from the cavity by adjusting the off-axis displacement of the concave mirror in [10] or the cylindrical lenses in [13]. However, the modes of the generated HG beam are discontinuous (i.e., the index $m$ and $n$ are discontinuous) as reported, which may be caused by the limited displacement accuracy of the IME or the inherent defect of the cavity structure. An intra-cavity PZT deformable mirror is used as an IME to modulate the HG beam in a Z-shaped cavity, and the HG mode could be continuously tuned from $HG_{0,0}$ to $HG_{0,9}$ by changing the driving voltage of the deformable mirror [12]. In this method, the high-order HG mode beam could not be generated using the intra-cavity PZT deformable mirror, which may be caused by the limited modulation range of the deformable mirror or the inherent defect of the resonant cavity structure. A spatial light modulation is used as a modulation cavity mirror [14,15], and the loading pattern on the spatial light modulation could be adjusted by changing the driving voltages of each spatial light modulation pixel. The mode of the HG beam up to $HG_{50,50}$ could be generated by loading special phase patterns in [14]. However, researchers have to spend extensive experimental time to find the proper patterns for a given cavity in particular, as the spatial light modulation could produce numerous different phase patterns.

For these methods, the mode of the generated HG beam could be tuned by adjusting the value of the modulation parameters (i.e., the off-axis displacement in [10,13] or the driving voltage in [12,14,15]) of the IMEs. However, in these methods, the theoretical relationship between the output light field from the cavity and the modulation of the IMEs is not investigated, and thus the influence of cavity parameters (e.g., the modulation value and accuracy, the structure parameters) on the output HG mode is unrevealed. As a result, for a given resonant cavity, it is very difficult to anticipate the modes of the HG beam generated from the cavity based on the modulation of the IMEs. Additionally, for a target HG beam, the required modulation value and accuracy are unknown, and thus it is difficult to select an IME with suitable parameters. If the modulation value or the accuracy of the selected IME is not proper, the mode of the HG beam generated from the cavity may be discontinuous. In particular, researchers have to conduct extensive and recurring experiments to find the relationship for a given resonant cavity, and the process is time-consuming. Unfortunately, if the cavity structure changes, the acquired relationship from the experiment is unsuitable for the new cavity and the experiments have to be repeated to obtain the new relationship. Furthermore, if an unsuitable IME is used in a given resonant cavity, it is possible that target HG beams could not be obtained through a time-consuming experiment. Therefore, to avoid the aforementioned problems, it is necessary to establish the theoretical relationship between the HG beam modes and the modulation parameters of the IMEs. As the generated HG beam could be converted to the corresponding vortex beam by an extra-cavity AMC of the vortex laser, the relationship between the modes of the vortex beam and modulation parameters could be revealed.

In this paper, a mode modulation regulation acquisition (MORA) method is proposed to investigate the relationship between the modes of the vortex beam and the modulation parameters for a vortex laser. This paper is organized as follows. In Section 2, the configuration of a mode-tunable vortex laser and the principle of the MORA method to acquire the mode modulation regulation are illustrated. In Section 3, a simulation to investigate the theoretical relationship between the tunable vortex beam mode and the modulation parameters of the IMEs is carried out. In Section 4, an experiment is conducted to verify the validity of the MORA method. The experiment results agree well with the simulation, which indicates that the mode modulation relationship obtained in the simulation could be used as a practical guidance in the experiment.

## 2. Theoretical Analysis

### 2.1. Analysis Model of a Mode-Tunable Optical Vortex Laser

To analyze the theoretical relationship between the mode-tunable vortex beam and the modulation parameters of the IMEs, a mode-tunable vortex laser (Figure 1) is used as the analysis object, which is composed of a classic off-axis pumped laser, a vortex beam generation beamline, and a reference beam acquisition beamline. Parameters of the mode-tunable laser are listed in Table 1. As shown in Figure 1, the resonant cavity is a plano-concave straight cavity, which consists of an input couple (IC) mirror, a gain medium, and an output couple (*OC*) mirror. For the resonant cavity, the pumping beam is focused into the gain medium by the couple lenses, which is composed of lenses L1 and L2. In the resonant cavity, the high-order HG beam could be generated by off-axis pumping, which is realized by adjusting the off-axis displacement of the *OC* mirror.

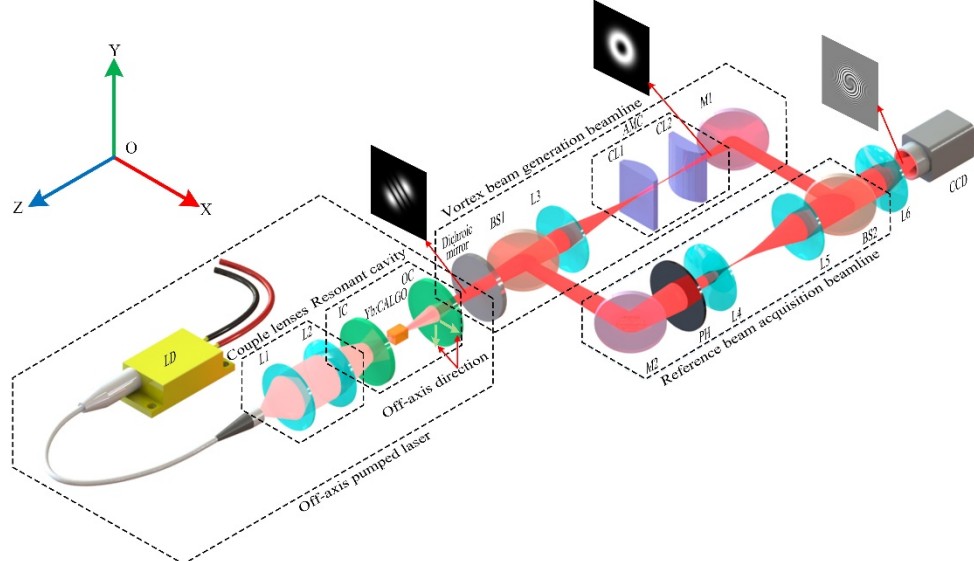

**Figure 1.** Configuration of the mode-tunable vortex laser.

**Table 1.** Parameters of the mode-tunable vortex laser.

| Parameter Symbol | Value | Meaning |
|---|---|---|
| $f_{L1}/f_{L2}/f_{L3}$ | 30 mm/60 mm/175 mm | Focal length of L1/L2/L3 |
| $f_{L4}/f_{L5}/f_{L6}$ | 125 mm/125 mm/30 mm | Focal length of L4/L5/L6 |
| $f_{CL1}/f_{CL2}$ | 25 mm/25 mm | Focal length of CL1/CL2 |
| $R_{OC}$ | 100 mm | Curvature radius of the *OC* mirror |
| $S_x/S_y/S_z$ | 2 mm/2 mm/4 mm | Length of the gain medium in the X/Y/Z-axis direction |
| $d$ | 6 mm | Distance from the IC to the gain medium |
| $L$ | 96 mm | Length of the resonant cavity |

As shown in Figure 1, the vortex beam generation beamline contains a dichroic mirror, a beam splitter BS1, a matching lens L3, an AMC, and a reflective mirror M1. The dichroic mirror is used to filter the residual pumping beam to avoid the influence of the pumping beam on the output beam. The BS1 is used to split part of the HG beam into the reference beam acquisition beamline. Lens L3 is used to transform the HG beam to match the conversion condition of the AMC. Composed of two identical cylindrical lenses (CL1 and CL2), the AMC could convert the incident HG beam to the corresponding vortex beam, which is collected by CCD after the reflection of M1, the reflection of BS2, and the focus of lens L6. The reference beam acquisition beamline is composed of a reflective mirror M2, a pinhole PH, lenses L4 and L5, as well as a beam splitter BS2. The PH is used to select part of the incident HG beam, which is expanded by a telescope consisting of lenses L4 and

L5. If lenses L4 and L5 are non-confocal, a spherical reference wave will be obtained. The BS2 is used to realize the interference between the vortex beam and the spherical reference wave, and the interference pattern is recorded by CCD. Lens L6 is used to adjust the size of the interference pattern on the CCD.

In Figure 1, the *OC* mirror is set as the IME, and thus the off-axis pumping of the resonant cavity could be achieved by adjusting the off-axis displacement of the *OC* mirror. If the off-axis displacement is suitable, a high-order HG beam will be generated from the cavity and the corresponding vortex beam will be obtained after passing through the vortex beam generation beamline.

### 2.2. The Mode Modulation Regulation Acquisition Method

Based on the theoretical model of the mode-tunable vortex laser, a MORA method is presented to investigate the theoretical relationship between the tunable vortex beam mode and the modulation parameters of the IMEs. The MORA method consists of three major processes, including the resonant cavity equivalent, intra-cavity iteration and light field output, as well as vortex beam generation and identification.

#### 2.2.1. Resonant Cavity Equivalent

To facilitate the establishment of a resonant cavity simulation model, the resonant cavity equivalent is first carried out to obtain the equivalent straight cavity for a given resonant cavity. For a non-straight cavity, a cavity expansion equivalent method is presented to obtain the equivalent straight cavity, which generally includes the cavity expansion process of equating the non-straight cavity to the straight cavity, and the phase equivalent process of calculating the additional phase of the optical elements.

For instance, Figure 2 shows the cavity expansion process of a LD pumped laser with a classic Z-shaped cavity (Figure 2a), which consists of an IC mirror, two reflection mirrors (RM1 and RM2), and an *OC* mirror. Here, the ideal planes TRP1 and TRP2 represent the theoretical reflection plane (TRP) of RM1 and RM2, respectively. In the cavity expansion process, the Z-shaped cavity could be expanded along the optical route to obtain an equivalent straight cavity, while the relationship between the optical route and the reflection mirrors remains the same. Following the cavity expansion process, an equivalent straight cavity of the Z-shaped cavity is achieved as shown in Figure 2b. Here, the *X–Y* coordinate shown in Figure 2b is the resonant cavity coordinate, and the *X′–Z′* coordinate shown in Figure 2b is the TRP1 coordinate of RM1.

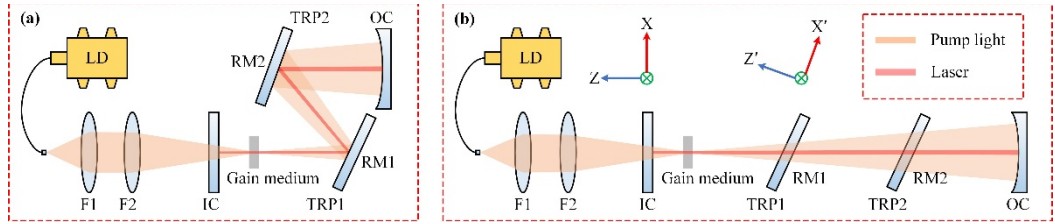

**Figure 2.** (**a**) Classic Z-shaped cavity; (**b**) the equivalent straight cavity after the cavity expansion process.

As well known, the light field propagation at the optical element in resonant cavity could be expressed as Equation (1).

$$E^{out}(x,y) = E^{in}(x,y)Am(x,y)exp[j \cdot IPF(x,y)], \tag{1}$$

where $E^{in}(x,y)$ and $E^{out}(x,y)$ represent the incident light field and the output light field, respectively. The $IPF(x,y)$ is the intra-cavity phase function of each optical element, which represents the additional phase of the optical element and the phase modulation on the light field. The $Am(x,y)$ is the intra-cavity amplitude function of each optical element, which represents the amplitude modulation on the light field.

In the equivalent straight cavity, as the original reflection mirrors are transformed to the new transparent elements, the IPF should be re-calculated in the phase equivalent process. Figure 3a shows the schematic diagram of the light field propagation on the RM1 in the *XZ*-section of the equivalent straight cavity. The horizontal pink arrows represent the propagating light field. $S_1$ represents the reflection surface shape of RM1, while $S_1$ and $S_1'$ are symmetric with respect to the ideal plane TRP1 based on the cavity expansion process. Figure 3b shows the resonant cavity coordinate *XYZ* and the TRP1 coordinate $X'Y'Z'$ of RM1. As shown in Figure 3b, $Y'(Y)$ axis is the axis of the *XOZ* coordinate and the $X'O'Z'$ coordinate, and $\theta$ represents the rotation angle. Of note, the $\theta$ is negative if the rotation direction from the *XOZ* coordinate to $X'O'Z'$ coordinate is clockwise, and $\theta$ is also the angle between the TRP1 and the *XOY* plane.

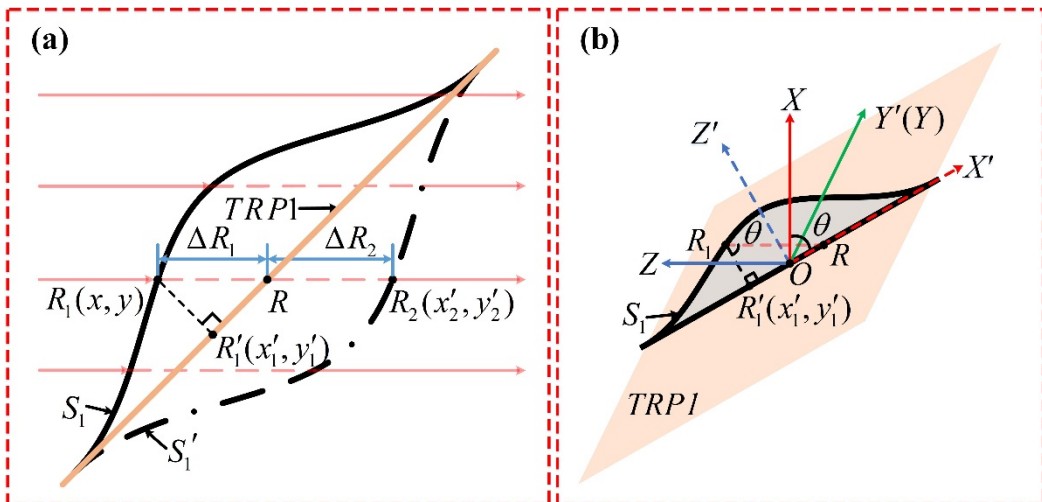

**Figure 3.** (**a**) The *XZ*-section of RM1; (**b**) the coordinate of the TRP1 and cavity.

As shown in Figure 3a, the IPF of the RM1 could be expressed as Equation (2).

$$IPF(x,y) = -k[\Delta R_1(x,y) + \Delta R_2(x,y)], \tag{2}$$

where $\Delta R_1(x,y)$ and $\Delta R_2(x,y)$ represent the additional optical paths induced by the reflection surface shape $S_1$ of RM1 and the symmetrical surface shape $S_1'$ of $S_1$, respectively. In addition, $k$ is the wave number of the light field. As shown in Figure 3a, point $R_1'$ is the vertical projection of point $R_1$ on the TRP1. In the TRP1 coordinate $X'Y'Z'$, the reflection surface shape of RM1 could be expressed as $f(x_1', y_1')$, and thus $|R_1 R_1'|$ in Figure 3 is equal to $f(x_1', y_1')$. The coordinate of point $R_1$ and the additional optical paths $\Delta R_1(x,y)$ in Figure 3a satisfy Equation (3), according to the geometric relationship.

$$\begin{cases} x = \left[x_1' - f(x_1', y_1')\tan\theta\right]\cos\theta \\ \quad\quad y = y_1' \\ \Delta R_1(x,y) = f(x_1', y_1')/\cos\theta \end{cases}, \tag{3}$$

As $S_1$ and $S_1'$ are symmetric with respect to the TRP1, $S_1'$ could be expressed as $S_1' = -f(x\prime, y\prime)$. According to the process of $\Delta R_1(x,y)$ calculation, the additional optical path $\Delta R_2(x,y)$ could be expressed as Equation (4).

$$\begin{cases} x = [x_2' - f(x_2', y_2')\tan\theta]\cos\theta \\ \quad\quad y = y_2' \\ \Delta R_2(x,y) = f(x_2', y_2')/\cos\theta \end{cases}, \tag{4}$$

Therefore, the IPF of RM1 could be obtained by Equations (2)–(4) based on the reflection surface shape $S_1$ in the TRP1 coordinate and the rotation angle $\theta$ of the TRP1. Similar to RM1, the IPFs of other optical elements (RM2, IC, and *OC*) in the equivalent straight cavity shown in Figure 2b, could also be calculated on the basis of the reflection surface shape of each element and the rotation angle $\theta$ of the corresponding TRP. In particular, for the IC and *OC* mirrors, the rotation angle $\theta$ of the IC and *OC* mirrors in the *XOY* coordinate is equal to 0. According to Equations (2)–(4), the IPFs, as well as the additional optical paths $\Delta R_1(x, y)$ and $\Delta R_2(x, y)$ of the IC and *OC* mirrors, could be calculated by Equation (5).

$$\begin{cases} \Delta R_1(x, y) = f(x, y) \\ \Delta R_2(x, y) = f(x, y) \\ IPF(x, y) = -2k \cdot f(x, y) \end{cases}, \tag{5}$$

where $f(x, y)$ is the reflection surface shape of the IC or *OC* mirrors in the cavity coordinate *XOY*.

### 2.2.2. Intra-Cavity Iteration and Light Field Output

The process of the intra-cavity iteration and light field output includes initial light field setting, intra-cavity light field propagation, iteration criterion, and light field output. The initial light field is set as a random distribution field with random amplitude distribution and random phase distribution, and could be expressed as Equation (6).

$$E(x, y) = A(x, y) exp[j \cdot B(x, y)], \tag{6}$$

where $A(x, y)$ and $B(x, y)$ represent the random amplitude distribution and random phase distribution, respectively.

In the equivalent straight cavity, the light field propagates contain the propagation in the air, the propagation in the gain medium, and the reflection by the optical mirrors. It is well known that the light propagation in air could be simulated by the Fresnel diffraction integral method. According to Equation (1), the influence of the gain medium on the light field consists of the amplitude modulation and phase modulation. In the analysis, the surfaces of gain medium are set as ideal planes, and thus the IPF is considered as equal to zero and the phase modulation could be ignored. The amplitude modulation $Am_g(x, y)$ is determined by the gain distribution of the gain medium, which could be expressed as Equation (7).

$$g(x, y) = g_0(x, y) / \left[ 1 + \frac{I(x, y)}{I_{sat}(x, y)} \right], \tag{7}$$

where $g_0(x, y)$ is the small signal gain distribution of the gain medium, which represents the initial gain distribution of the gain medium. $g(x, y)$ represents the gain distribution when the gain saturation is considered. $I(x, y)$ represents the light intensity distribution in the gain medium, while $I_{sat}(x, y)$ is the saturation intensity of the gain medium. The amplitude modulation $Am_g(x, y)$ could be expressed as Equation (8).

$$Am_g(x, y) = exp \left[ \frac{1}{2} g(x, y) S_z \right], \tag{8}$$

where $S_z$ represents the length of the gain medium in the *Z*-axis. The light field propagation at optical mirrors could be calculated by Equation (1). The amplitude modulation $Am(x, y)$ is equal to the transmission ratio or the reflectivity of each optical element, determined by the light field passing through or reflected by the optical element. The phase modulation $IPF(x, y)$ of each optical element could be calculated by Equations (2)–(4) in the phase equivalent process.

Figure 4 shows the light field propagation in the equivalent straight cavity in Figure 2a. As shown in Figure 4, the light field propagation could be divided into three

types, including the propagation in the air (②/④/⑥/⑧/⑩/⑫/⑭/⑯), the reflection by the optical mirrors (①/③/⑤/⑨/⑬/⑮), and the propagation in the gain medium (⑦/⑪). The propagation of different types could be calculated by Equation (9).

$$\begin{cases} P_{air} : E^{out}(x,y) = \frac{exp(jkz)}{j\lambda z} \iint E^{in}(x_1,y_1)exp\left\{\frac{jk}{2z}\left[(x-x_1)^2 + (y-y_1)^2\right]\right\}dx_1dy_1 \\ P_{ref} : E^{out}(x,y) = RE^{in}(x,y)exp[j\cdot IPF(x,y)] \\ P_{cry} : E^{out}(x,y) = Am_g E^{in}(x,y) \end{cases} \quad , \quad (9)$$

where $P_{air}$, $P_{ref}$, and $P_{cry}$ represent the propagation in the air, the reflection by the optical mirrors, and the propagation in the gain medium, respectively. $E^{in}(x,y)$ and $E^{out}(x,y)$ represent the input light field and the output light field after each propagation, respectively. $k$ and $\lambda$ represent the wave number and the wave length of the light field, respectively, while $z$ represents the propagation distance in the air. $(x_1,y_1)$ represents the coordinate in the input plane, and $R$ represents the reflectivity of the mirror. $IPF(x,y)$ represents the additional phase of the mirror, which could be calculated by the phase equivalent method. $Am_g(x,y)$ represents the amplitude modulation of the gain medium, which could be calculated by Equation (8). Given that the initial light field $E_i(i = 0)$ starts from the *OC* mirror, it will be changed to $E_{i+1}$ after propagating sequentially from ① to ⑯ in the cavity, which is called the intra-cavity propagation iteration and could be calculated by Equation (10).

$$\begin{cases} H_v(x,y) = E_v^{out}(x,y)/E_v^{in}(x,y) \\ E_{i+1}(x,y) = \prod_{v=1}^{Nv} H_v(x,y)\cdot E_i(x,y) \end{cases} \quad , \quad (10)$$

where $Nv$ represents the number of the propagation stage of one intra-cavity propagation iteration (e.g., $Nv = 16$ in Figure 4), and $i$ represents the number of iterations. $H_v(x,y)$ represents the modulation on the light field in the $v$th propagation stage (e.g., passing through the air, the mirror or the gain medium). When the iteration threshold value is met and the stable condition in Equation (11) is satisfied, the light field will tend to be stable and output from the cavity.

$$max\{||E_{i+1}(x,y)| - |E_i(x,y)||\} < \varepsilon, \quad (11)$$

where $\varepsilon$ is the threshold value. $E_i(x,y)$ is the incident light field on the *OC* mirror after the $i$th intra-cavity propagation iteration.

### 2.2.3. Vortex Beam Generation and Identification

As shown in Figure 1, the output laser beam from the resonant cavity is converted to the vortex beam ($LG_{p,l}$ beam) by an extra-cavity AMC in the vortex beam generation beamline. As the mode of the vortex beam is represented by the radius parameter $p$ and azimuthal parameter $l$, the vortex beam identification contains the parameter $p$ identification and the parameter $l$ identification, which is realized by the recognition on the intensity distribution of the vortex beam and on the interference pattern between the vortex beam and a spherical wave, respectively. The values of the parameters $p$ and $l$ could be calculated by Equation (12).

$$\begin{cases} p = N_r - 1 \\ |p - l| = N_s \end{cases} \quad , \quad (12)$$

where $N_r$ is the number of the intensity rings in the intensity distribution and $N_s$ is the number of the spiral fringes in the interference pattern. For instance, according to the intensity of a vortex beam and the corresponding interference pattern shown in Figure 5, the value of $N_r$ is equal to 1 (Figure 5a) and the value of $N_s$ is equal to 3 (Figure 5b). Based on Equation (12), the value of parameter $p$ is equal to 0, and the value of parameter $l$ is equal to 3, if the sign of $l$ is ignored. Of note, if both parameters $N_r$ and $N_s$ are equal to

zero, this indicates that the output laser beam from the resonant cavity is not an HG beam and the converted beam by the AMC is not a vortex beam.

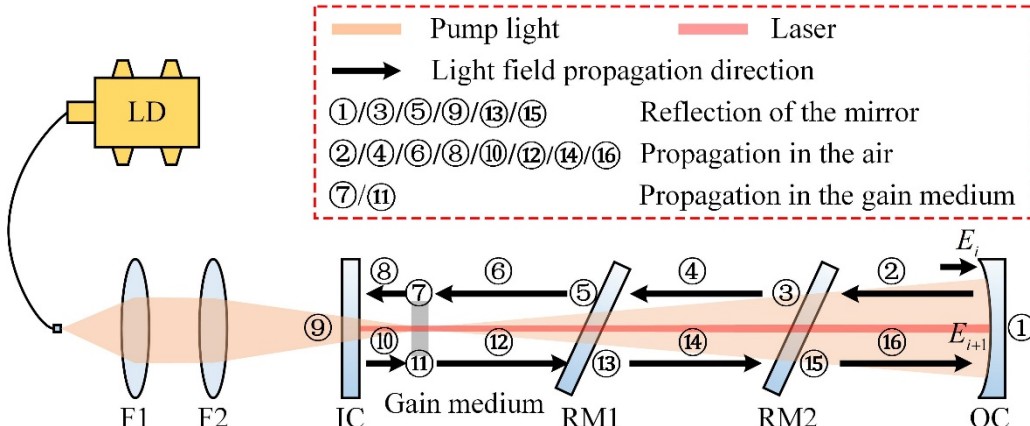

**Figure 4.** Schematic of the intra-cavity propagation iteration of the light field.

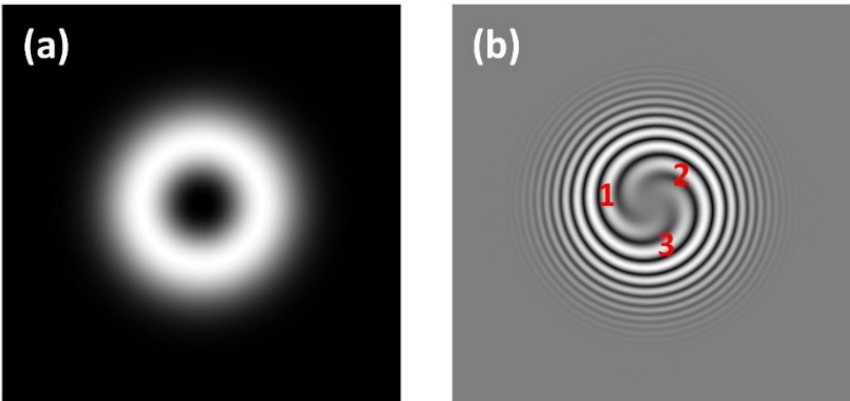

**Figure 5.** (**a**) Intensity distribution of a vortex beam; (**b**) interference pattern of the vortex beam and a spherical wave.

### 2.2.4. Mode Modulation Regulation Acquisition

Figure 6 shows the mode modulation regulation acquisition method, including the three major processes illustrated above. The symbols and the corresponding meanings in Figure 6 are listed in Table 2. Based on the MORA method, the theoretical relationship between the tunable vortex beam mode and the modulation parameters of the IME could be achieved. As illustrated above and shown in Figure 6, the process to obtain the theoretical relationship could be divided into three steps, including the resonant cavity equivalent, the intra-cavity and light field output, as well as vortex beam generation and identification.

**Table 2.** The symbols and the corresponding meanings.

| Symbol | Meaning |
|--------|---------|
| MPV | Modulation parameter value of the IME |
| $\Delta$ | Increment of the MPV |
| MV | Maximum value of the MPV |
| $\varepsilon$ | Criterion value of the intra-cavity light field propagation |

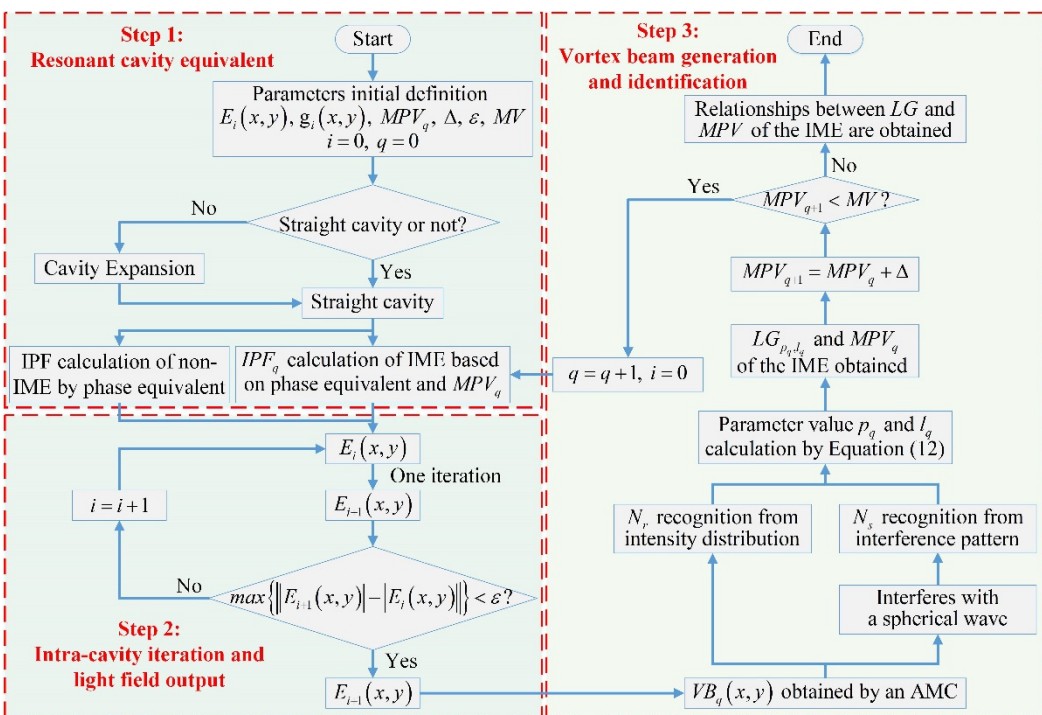

**Figure 6.** Working process of the MORA method.

In step 1, primary parameters of the resonant cavity will be defined first, including the initial light field $[E_0(x,y)]$ on the *OC* mirror, the initial gain distribution $[g_0(x,y)]$ of the gain medium, the initial modulation parameter value ($MPV_0$) of the IME, the increment ($\Delta$) of the modulation parameter of the IME, the criterion value ($\varepsilon$) of the cavity, and the maximum value (MV) of the MPV. Then, the resonant cavity equivalent is carried out to identify the cavity type and the non-straight cavity is converted to a straight cavity by the cavity expansion process. In the equivalent straight cavity, the IPFs of the non-IME and the IMEs are calculated based on the $MPV_j$ by the phase equivalent process.

In steps 2 and 3, the incident light fields $E_i(x,y)$ and $E_{i+1}(x,y)$ on the *OC* mirror after the $i^{th}$ and $(i+1)$th intra-cavity propagation iterations are calculated. When the iteration threshold value is met, and the light field is considered as stable and set as output from the cavity, it will be converted by an AMC to a new laser beam $VB(x,y)$. Based on the intensity of the laser beam $VB(x,y)$ and the interference pattern between the laser beam $VB(x,y)$ and spherical wave, the values of parameters $N_r$ and $N_s$ could be recognized. According to Equation (12), the mode (i.e., the values of $p$ and $l$) of the laser beam could be identified. If the laser beam is confirmed as a vortex beam, the theoretical relationship between the recognized $LG_{p,l}$ beam and the present $MPV_q$ of the IME in the mode-tunable vortex laser is obtained. Otherwise, a new MPV will be generated and the corresponding IPF of the IME will be calculated. Based on the new IPF, steps 2 and 3 will be repeated to identify the radius parameter $p$ and the azimuthal parameter $l$ of the laser beam $VB(x,y)$. Therefore, within the modulation range $[MPV_0, MV]$ of the IME, the theoretical relationship between all of the available vortex beams and the modulation parameters of the IME in the mode-tunable vortex laser is finally obtained. Of note, if no vortex beam could be recognized during the whole process, the laser system will be considered as having no capability to generate a vortex beam and the configuration or elements of the cavity should be re-designed.

### 3. Simulation

#### 3.1. Simulation Model

Based on the theoretical model of the mode-tunable vortex laser and the MORA method, the theoretical relationship between the tunable vortex beam mode and the modulation parameters of the IMEs is investigated in the simulation. The structure and the parameters of the vortex laser are shown in Figure 1 and Table 1, respectively. In the resonant cavity part of the laser, the *OC* mirror is set as the IME and could be moved along the *X*- and *Y*-axis to realize off-axis pumping and achieve the high-order HG beam. Figure 7 shows the off-axis schematic diagram of the *OC* mirror, where *XOY* and *X'O'Y'* represent the coordinates of the resonant cavity and the *OC* mirror, respectively. Point *O'* is the center point of the *OC* mirror. In addition, $(dx, dy)$ are the modulation parameters and represent the off-axis displacement of the *OC* mirror in the *X*- and *Y*-axis directions, respectively.

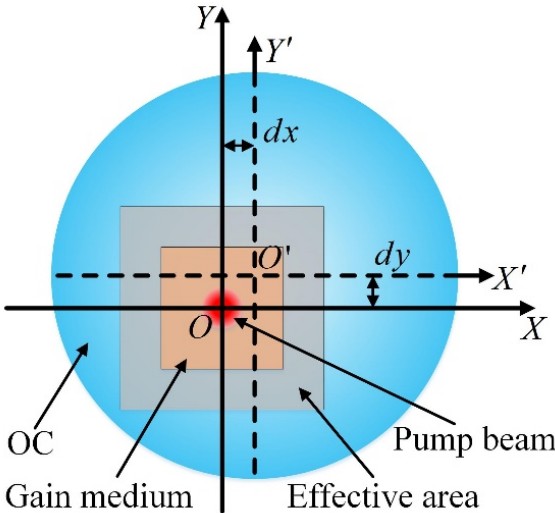

**Figure 7.** Schematic diagram of the off-axis *OC* mirror.

In the simulation, the resonant cavity is set as a classic plano-concave straight cavity and the initial light field $E_0(x, y)$ on the *OC* mirror is defined as $A(x, y) \in (0, 1)$ and $B(x, y) \in (0, 2\pi)$ (Equation (6)). The initial gain distribution $g_0(x, y)$ of the gain medium is set as a Gaussian distribution with a waist radius of 200 μm. The surfaces of the IC mirror and the gain medium are set as ideal planes and the IPFs are defined as equal to zero. The initial off-axis displacements $(dx, dy)$ are defined as $(0, 0)$. The threshold value $\varepsilon$ of the criterion is set as $10^{-3}$. To avoid a serious diffraction caused by the gain medium, the maximum value MV of the off-axis displacements $(dx, dy)$ is defined as 350 μm. The surface shape of the effective area (grey square area in Figure 7, side length 6 mm) of the *OC* mirror will be changed by adjusting the off-axis displacements $(dx, dy)$, and the IPF of the *OC* within the effective area could be expressed as Equation (13).

$$IPF_{oc}(dx, dy) = 2k\frac{(x - dx)^2 + (y - dy)^2}{2R_{oc}}, \tag{13}$$

where $R_{OC}$ is the curvature radius of the reflection surface of the *OC* mirror.

Based on the initial values of the parameters, an intra-cavity propagation iteration is carried out and the light field will output from the cavity until the iteration threshold value is met. The generated $E_{i+1}(x, y)$ laser beam is converted to a new laser beam $VB(x, y)$ using a cylinder lens transformation algorithm. Following this step, a recognition algorithm will be carried out to identify the values of parameters $N_r$ and $N_s$ and the mode of the laser beam. In the simulation, the *OC* mirror will be moved away from its original position and the IPF of the *OC* mirror will be updated to obtain new laser beams. Therefore, according

to Equation (13), the influence of the off-axis displacements $(dx, dy)$ on the output laser beam $VB(x, y)$ could be obtained.

### 3.2. Simulation Results

In the simulation, the vortex beam generation abilities with single off-axis and dual off-axis displacements are investigated. According to the simulation results, the modes of the tuned laser beam based on the IPF modulation could be obtained for a given cavity within a modulation range. The influence of the modulation accuracy of the selected IME (i.e., the *OC* mirror) on the modes of the generated laser beam is studied, and the stable region of the tuned mode is illustrated. Based on these results, the theoretical relationship between the modes of the vortex beam and modulation parameters could be revealed.

### 3.2.1. Vortex Beam Generation Ability with Single Off-Axis Displacement

In this simulation section, the single off-axis displacement is realized by moving the *OC* mirror along the *X*-axis. Therefore, the deviation $dx$ from the original position (Figure 7) is the modulation parameter, while no off-axis displacement occurs in *Y*-axis and $dy$ is equal to 0. The increment value $\Delta$ of the modulation parameter is set as 5 μm. According to the working process of the MORA method, a simulation is carried out and the relationship between the off-axis displacement value $dx$ and the modes of the generated laser beam could be obtained.

In Figure 8, the 1st and 3rd rows represent the HG beam generated from the resonant cavity after intra-cavity propagation iterations with different off-axis displacement values $dx$. The 2nd and 4th rows represent the interference pattern of a spherical wave interfering with the vortex beam, which is converted from the HG beam in the 1st and 3rd rows, respectively. As shown in Figure 8, a fundamental mode $(HG_{0,0})$ is generated when no off-axis displacement occurs and $dx$ is equal to 0. As shown in the 1st and 3rd rows, it could be seen that the mode of the generated HG beam could be continuously tuned from $HG_{0,0}$ to $HG_{14,0}$ beam within the modulation range [0, 350 μm]. Then, the HG beam is converted by an AMC, which consists of two cylindrical lenses. The vortex beams from $LG_{0,1}$ mode with $1\hbar$ OAM to $LG_{0,14}$ mode with $14\hbar$ OAM are obtained as shown in the 2nd and 4th rows. Figure 9a,b shows the enlarged views of the interference patterns marked in red dotted line in Figure 8, and the spiral fringes could be clearly identified. Of note, the mode of the converted vortex beam is identified from the values of parameters $N_r$ and $N_s$. According to the simulation results, the increment value $\Delta$ of the modulation parameter should be set as precise enough (e.g., 5 μm or 1 μm) to ensure that no mode is missing. For instance, if the increment value $\Delta$ is set as large as 200 μm, the majority of the vortex beam modes could not be obtained from the mode-tunable vortex laser.

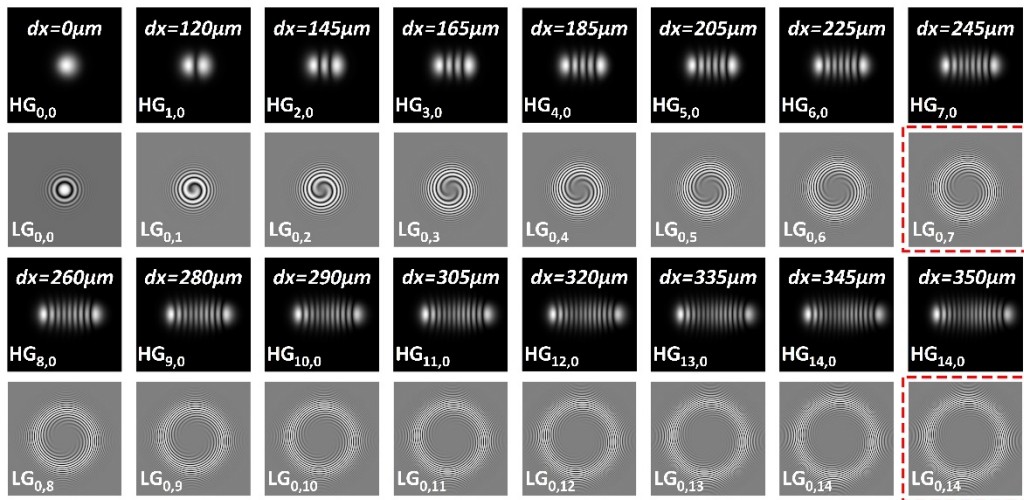

**Figure 8.** Generation of HG and vortex beams with single off-axis displacement.

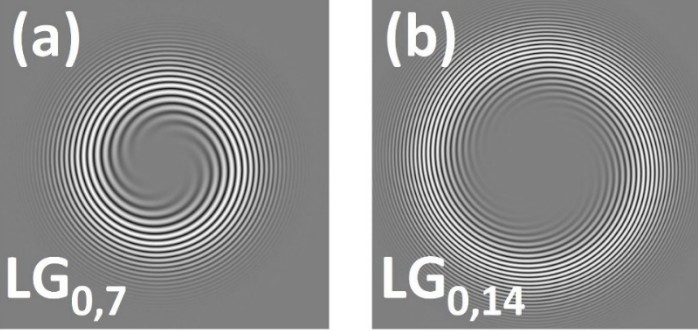

**Figure 9.** Enlarged views of the interference patterns marked in red dotted line in Figure 8, (**a**) $LG_{0,17}$ and (**b**) $LG_{0,14}$.

Figure 10a,b shows the output field of the cavity with the off-axis displacements $dx = 500$ and 550 μm, respectively. From Figure 10a, it could be seen that the high-order HG beam is generated, but the diffraction appears in the green dotted line and the light spots are blurred in the red dotted line. With the increase in the off-axis displacement, the diffraction becomes more serious and the laser beam output from the cavity is no longer an HG mode beam as shown in Figure 10b. Therefore, to avoid the influence of the diffraction on the output laser beam, the off-axis displacement of the *OC* mirror should be set within a suitable modulation range (e.g., set within 350 μm in our simulation and experiment).

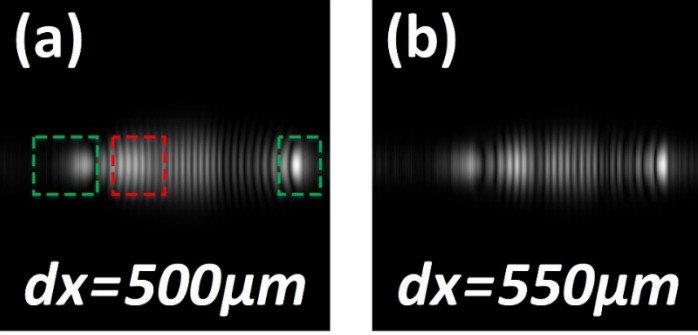

**Figure 10.** The output light field of the cavity with (**a**) $dx = 500$ μm and (**b**) $dx = 550$ μm, respectively.

### 3.2.2. Vortex Beam Generation Ability with Dual Off-Axis Displacement

To analyze the influence of the off-axis displacement in both *X*- and *Y*-axis, the vortex beam generation ability with dual off-axis displacement is investigated in this section. The dual off-axis displacement is realized by moving the *OC* mirror along the *X*- and *Y*-axis. Therefore, the deviation $(dx, dy)$ from the original position (Figure 7) is the modulation parameter. The increment value $\Delta$ of the modulation parameter is also set as 5 μm. In the simulation, to conveniently describe the dual off-axis displacement of the *OC* mirror, parameters $dr$ and $\theta_r$ are introduced as shown in Equation (14).

$$\begin{cases} dr = \sqrt{dx^2 + dy^2} \\ \theta_r = arctan(dy/dx) \end{cases} \tag{14}$$

where $\theta_r$ represents the off-axis direction of the *OC* mirror in the *XOY* coordinate, and $dr$ represents the off-axis displacement along the direction. In the simulation, the $\theta_r$ direction is first set and the relationship between the off-axis displacement value $dr$ and the modes of the generated laser beam along the $\theta_r$ direction could be obtained. Figure 11 shows some typical simulation results for different off-axis displacement values $dr$ (i.e., $dr = 120$ μm, $dr = 165$ μm, $dr = 205$ μm, $dr = 260$ μm, $dr = 290$ μm, $dr = 320$ μm, and $dr = 335$ μm) in four off-axis directions (i.e., $\theta_r = 0$, $\theta_r = \pi/4$, $\theta_r = \pi/3$, and $\theta_r = \pi/2$).

In Figure 11, the 1st, 3rd, 5th, and 7th rows represent the HG beam generated from the resonant cavity after intra-cavity propagation iterations with different off-axis displacement values $dr$ along the off-axis direction. The 2nd, 4th, 6th, and 8th rows represent the interference pattern of a spherical wave interfering with the vortex beam, which is converted from the HG beam. In the first column of Figure 11, the white arrows in the odd rows represent the off-axis direction of the $OC$ mirror. In each odd row, it could be seen that the laser beam splits along the off-axis direction, and the mode order of the laser beam increases with the increment of the off-axis displacement value $dr$. Of note, in each column, the modes of the generated HG and converted vortex beams are identified as the same in different off-axis directions $\theta_r$, as the off-axis displacement value $dr$ is the same. Based on the simulation results, the theoretical relationship between the modulation parameter, i.e., $(dr, \theta_r)$ or $(dx, dy)$, and the modes of the vortex beam is obtained.

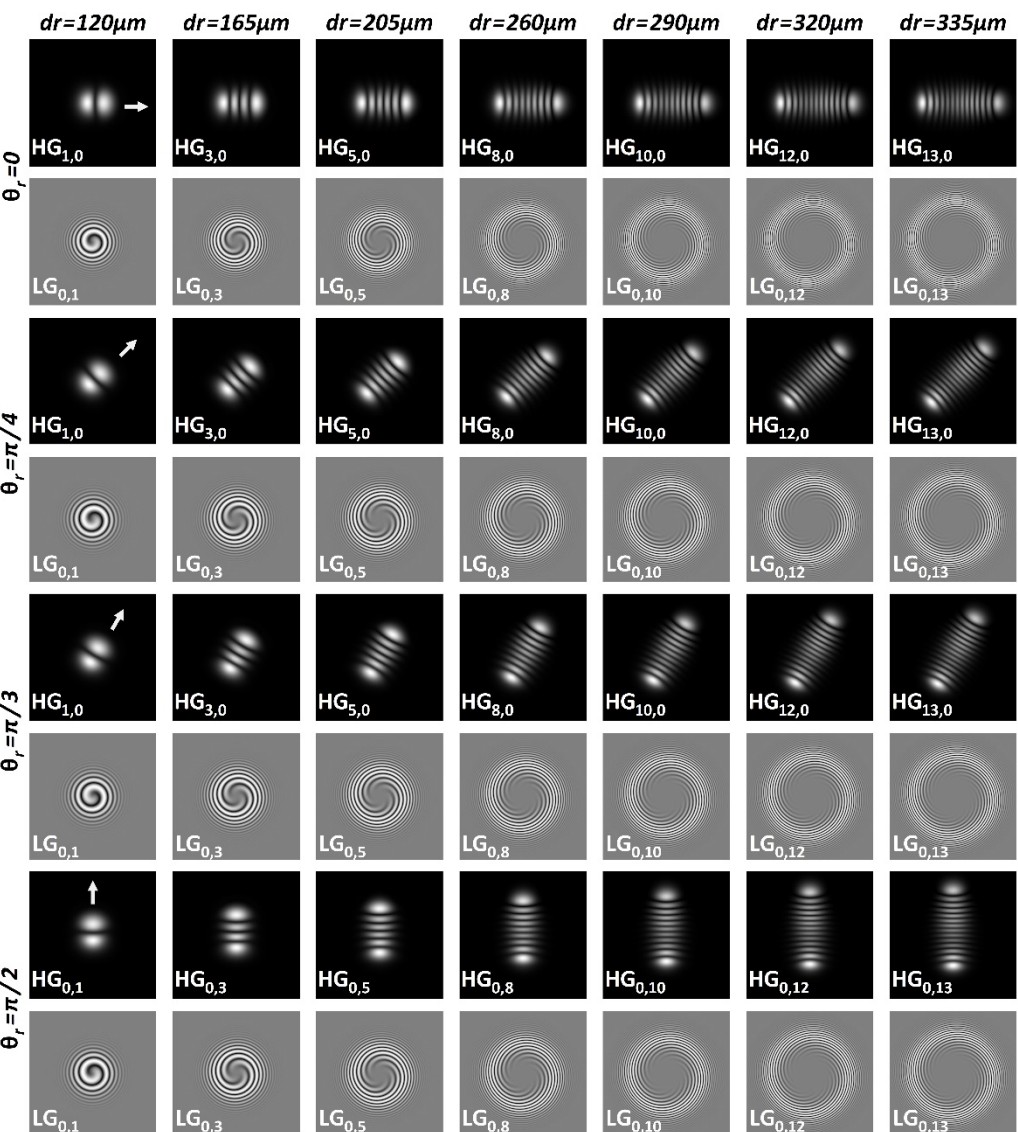

**Figure 11.** Generation of HG and vortex beams with dual off-axis displacement.

### 3.2.3. Stable Region for the Tuned Beam Mode

To analyze the influence of the increment value (i.e., modulation accuracy) on the generated beam mode, the stable region for the tuned mode is investigated in this section. In the simulation, the single off-axis displacement along the $X$-axis is considered for simplicity and the deviation $dx$ from the original position is considered as the modulation parameter.

The $HG_{1,0}$ beam is regarded as the target tuned beam and the increment value $\Delta$ of the modulation parameter is set as 0.2 μm. Figure 12 shows the simulation results of the evolution process from $HG_{0,0}$ to $HG_{2,0}$ beam and the stable region for the $HG_{1,0}$ beam.

In Figure 12, the 1st and 2nd rows represent the HG beam generated from the resonant cavity after intra-cavity propagation iterations with different off-axis displacement values $dx$. The 3rd row represents the interference pattern of a spherical wave and the vortex beam converted from the HG beam shown in the 2nd row. As shown in the 2nd and 3rd rows, within the whole modulation range [115 μm, 130 μm], the generated $HG_{1,0}$ beam is distinct and the converted laser beam by the AMC could be identified as the typical $LG_{0,1}$ vortex beam. When the off-axis displacement value $dx$ is set as 114 μm, the generated $HG_{1,0}$ beam is deteriorated, as the two split spots of the laser beam are blurred and diverse. When the off-axis displacement value $dx$ is set as 130.2 μm, a new spot emerges and the laser beam evolves from $HG_{1,0}$ to $HG_{2,0}$ beam. Therefore, the 15 μm modulation range of [115 μm, 130 μm] is identified as the stable region for the target $HG_{1,0}$ beam. Similarly, the stable regions for other HG beams could also be obtained. Based on the stable regions, a proper increment value of the modulation parameter could be selected to ensure the continuity of the generated vortex beam mode, which could be used as an important criterion to select the IME with a suitable modulation accuracy to generate the target beam in the experiment.

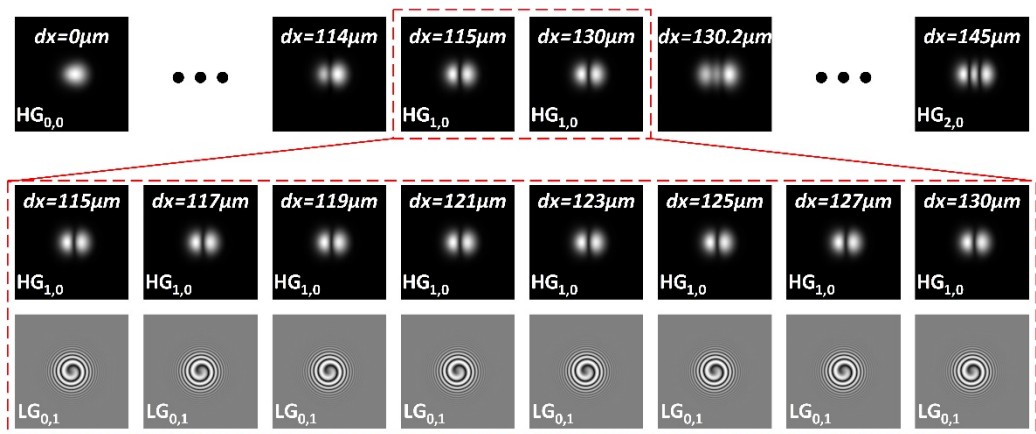

**Figure 12.** Stable region for the $HG_{1,0}$ beam.

## 4. Experiment Results

To investigate the validity of the MORA method, a mode-tunable vortex laser with primary parameters, which is set as the same with the simulation model (Table 1), has a similar configuration to Figure 1 and is built as shown in Figure 13. In the experiment setup, the classic plano-concave resonant cavity part is composed of an IC mirror (AR at 976 nm and HR at 1064 nm), an Yb:CALGO crystal (Altechna, 5 at%, a-cut, AR at 1064 nm), and a concave *OC* mirror (HR at 976 nm, transmittance is 2% at 1064 nm). The pump beam, generated by a fiber-coupled LD (20 W, 976 nm, 105 μm core diameter, 0.22 NA, BWT Beijing Ltd., Beijing, China), is focused into the gain medium (Yb:CALGO) with the beam waist radius of 200 μm by the couple lenses. In the vortex beam generation beamline, the HG beam passes through a dichroic mirror (HR at 976 nm and HT at 1064 nm), a beam splitter BS1 (transmittance is 10% at 1064 nm), a lens L3 (focal length 175 mm), and an AMC. Then, it is converted to the corresponding LG vortex beam. In the reference beam acquisition beamline, the laser beam, reflected by the reflective mirror M2 (HR at 1064 nm), passes through the pinhole PH (hole diameter 5 mm) and two spherical lenses (L4 and L5 set as non-confocal, focal length 25 and 125 mm), and is selected as a reference spherical wave. The reference spherical wave passes through the beam splitter BS2 and interferes with the vortex beam reflected by the reflective mirror M1 (HR at 1064 nm). The interference pattern is focused by lens L6 (focal length 30 mm) and recorded by a CCD (Point Grey, Grasshopper3 GS3-U3-14S5M).

In the experiment, the *OC* mirror, which is set as the IME in the cavity, will be moved away from its initial position and the IPF of the *OC* mirror will be changed to achieve the high-order HG beam under the off-axis pumping condition. The initial position of the *OC* mirror is set strictly in the co-axis with the cavity to generate the fundamental mode. The increment value Δ and the maximum value MV of the modulation parameter (i.e., the displacement of the *OC* mirror) are set as 1 and 350 μm, respectively, which is ensured by the adopted high-precision displacement stage.

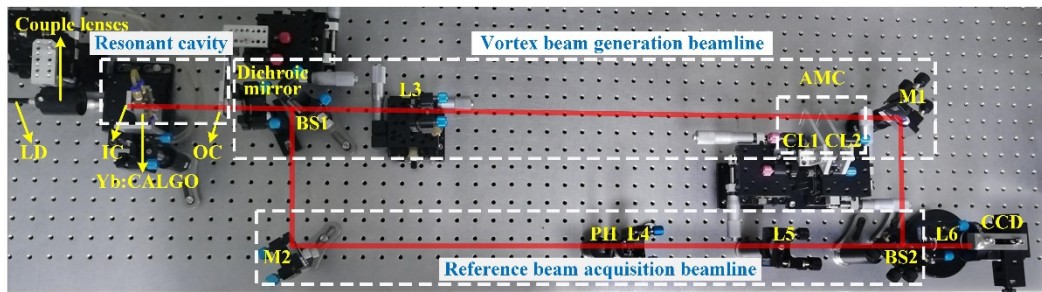

**Figure 13.** Experiment setup of the mode-tunable vortex beam.

Figure 14 shows the experiment results at the condition of the *OC* mirror off-axis, which is moved along the *X*-axis. The 1st and 3rd rows represent the HG beam generated from the resonant cavity, while the 2nd and 4th rows represent the interference pattern of a reference spherical wave interfering with the vortex beam that is converted from the HG beam in the 1st and 3rd rows, respectively. As shown in the 1st and 3rd rows, it could be seen that the mode of the generated HG beam could be continuously tuned from $HG_{0,0}$ to $HG_{14,0}$ within the displacement range [0, 350 μm]. Based on the HG beam, the vortex beams from $LG_{0,1}$ beam with 1ℏ OAM to $LG_{0,14}$ beam with 14ℏ OAM are obtained by an AMC and are shown in the 2nd and 4th rows. For each generated HG beam, the off-axis displacement value $dx$ in the experiment and simulation is listed in Table 3. It could be seen from Table 3 that the maximal difference $\Delta_{dx}$ between the experiment and simulation is as small as 5 μm, which indicates that the experiment results agree well with the simulation. Figure 15 shows three typical HG and LG beams, which are achieved in the experiment with different dual off-axis displacement values $(dx, dy)$. The white arrows represent the off-axis directions of the *OC* mirror in the mode tuning experiment, which are calculated based on the off-axis displacement values $(dx, dy)$ by Equation (14). It could be seen that the HG beam splits along the off-axis direction of the *OC* mirror, which is the same as the simulation results. The comparison of the off-axis displacement values $(dx, dy)$ in the simulation and experiment is shown in Table 3, where the maximal differences $\Delta_{dx}$ and $\Delta_{dy}$ are as small as 3 and 4 μm, respectively. The comparison in Table 3 indicates that the mode modulation regulation obtained in the simulation could be used as a practical guidance to realize the target mode of the HG and vortex beams in the experiment.

Figure 16 shows the experiment results of the evolution process from $HG_{0,0}$ to $HG_{2,0}$ beam and the stable region for the $HG_{1,0}$ beam. In Figure 16, the 1st row represents the HG beam, which is generated from the resonant cavity when the *OC* mirror is moved along the *X*-axis in the experiment. The 2nd row represents the interference pattern of a reference spherical wave and the vortex beam, which is converted from the HG beam as shown in the 1st row. Within the displacement range [114 μm, 133 μm], it could be seen that the generated $HG_{1,0}$ beam is clear and the spiral fringe identifying the vortex could be clearly observed in the interference pattern. When the off-axis displacement value $dx$ is adjusted to 113 μm, the two split spots of the laser beam are blurred and the generated $HG_{1,0}$ beam becomes deteriorated. The spiral fringe in the interference pattern is obscured and the values of the vortex parameters $N_r$ and $N_s$ could not be identified. When the off-axis displacement value $dx$ is adjusted to 132 μm, a new blurred spot emerges and the spiral fringe in the interference is also deteriorated. Of note, the laser beam evolves into the

$HG_{2,0}$ beam when the off-axis displacement value $dx$ is adjusted to 144 μm. The evolution process of the $HG_{1,0}$ beam in the experiment agrees well with the simulation results. As the increment value of the modulation parameter (i.e., displacement in the experiment) is selected as small as 1 μm, the continuity of the generated vortex beam mode is achieved and the stable region for a target vortex beam is obtained in the experiment.

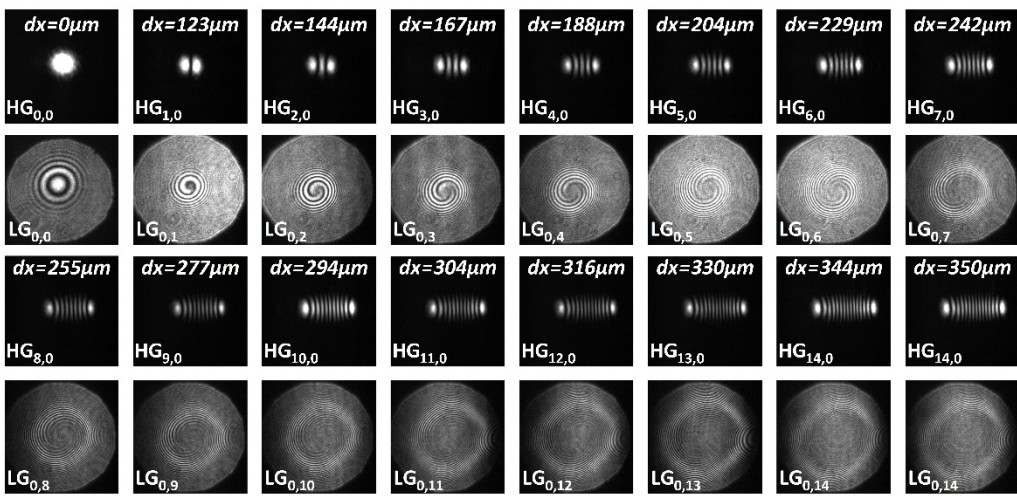

**Figure 14.** The generated HG and vortex beams in the single off-axis displacement experiment.

**Table 3.** The comparison of the off-axis displacements $(dx, dy)$ in the simulation and experiment.

| Off-Axis Method | HG Mode | $dx$/μm | | | $dy$/μm | | |
|---|---|---|---|---|---|---|---|
| | | Sim. | Exp. | $\Delta_{dx}$ | Sim. | Exp. | $\Delta_{dy}$ |
| Single off−axis | $HG_{1,0}$ | 120 | 123 | 3 | 0 | 0 | 0 |
| | $HG_{2,0}$ | 145 | 144 | −1 | 0 | 0 | 0 |
| | $HG_{3,0}$ | 165 | 167 | 2 | 0 | 0 | 0 |
| | $HG_{4,0}$ | 185 | 188 | 3 | 0 | 0 | 0 |
| | $HG_{5,0}$ | 205 | 204 | −1 | 0 | 0 | 0 |
| | $HG_{6,0}$ | 225 | 229 | 4 | 0 | 0 | 0 |
| | $HG_{7,0}$ | 245 | 242 | −3 | 0 | 0 | 0 |
| | $HG_{8,0}$ | 260 | 255 | −5 | 0 | 0 | 0 |
| | $HG_{9,0}$ | 280 | 277 | −3 | 0 | 0 | 0 |
| | $HG_{10,0}$ | 290 | 294 | 4 | 0 | 0 | 0 |
| | $HG_{11,0}$ | 305 | 304 | −1 | 0 | 0 | 0 |
| | $HG_{12,0}$ | 320 | 316 | −4 | 0 | 0 | 0 |
| | $HG_{13,0}$ | 335 | 330 | −5 | 0 | 0 | 0 |
| | $HG_{14,0}$ | 345 | 344 | −1 | 0 | 0 | 0 |
| Dual off−axis | $HG_{10,0}$ | 205 | 208 | 3 | 205 | 207 | 2 |
| | $HG_{9,0}$ | 130 | 128 | −2 | 225 | 229 | 4 |
| | $HG_{5,0}$ | 0 | 0 | 0 | 205 | 208 | 3 |

In the experiment setup, the pumping power is set as 10 W, which is the maximum output power of the LD. For all of the modes obtained in the vortex laser, the maximum output powers are tested and listed in Table 4. It could be seen that the maximum output power of the generated vortex laser decreases from 1.222 to 0.291 W with the mode order increasing from $LG_{0,1}$ to $LG_{0,14}$, while the optical−optical conversion efficiency decreases from 12.22% to 2.91%.

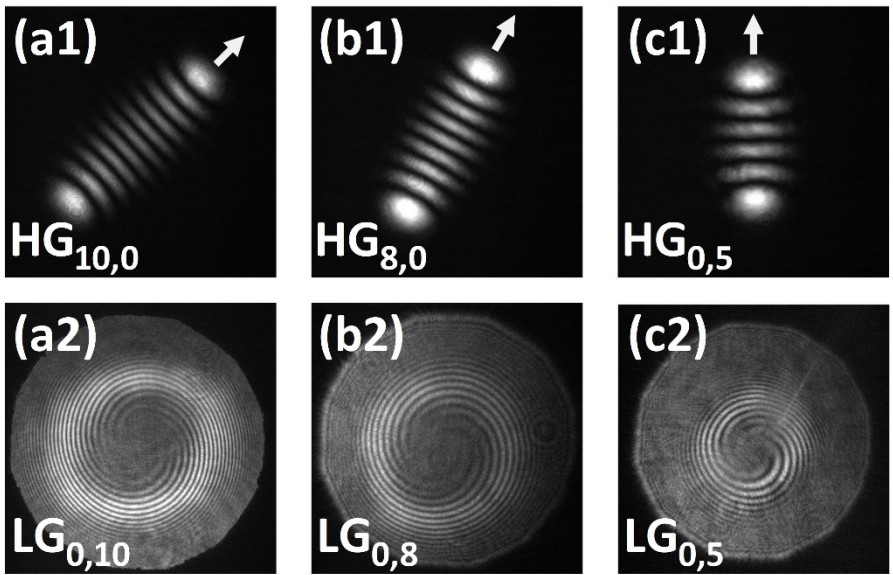

**Figure 15.** The generated HG and vortex beams in the dual off-axis displacement experiment with the displacement values $(dx, dy)$ set as (**a1**) and (**a2**) (208 μm, 207 μm), (**b1**) and (**b2**) (128 μm, 229 μm), (**c1**) and (**c2**) (0 μm, 108 μm).

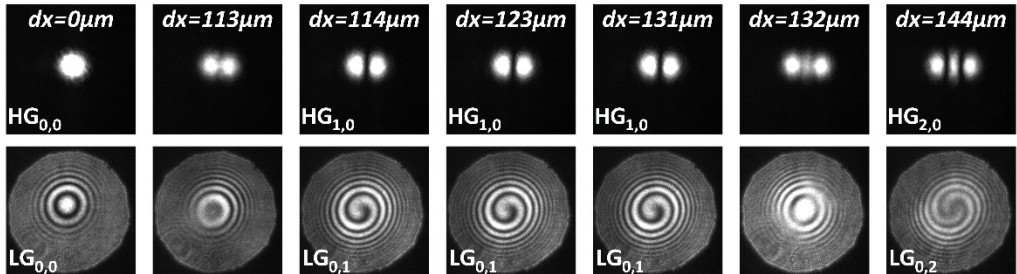

**Figure 16.** Experiment results of the evolution process from $HG_{0,0}$ to $HG_{2,0}$ beam and the stable region for the $HG_{1,0}$ beam.

**Table 4.** The maximum output power for each vortex mode of single off-axis displacement.

| Beam mode | $LG_{0,1}$ | $LG_{0,2}$ | $LG_{0,3}$ | $LG_{0,4}$ | $LG_{0,5}$ | $LG_{0,6}$ | $LG_{0,7}$ |
|---|---|---|---|---|---|---|---|
| Maximum output power (W) | 1.222 | 1.033 | 0.883 | 0.774 | 0.729 | 0.646 | 0.607 |
| Optical−optical Efficiency (%) | 12.22 | 10.33 | 8.83 | 7.74 | 7.29 | 6.46 | 6.07 |
| Beam mode | $LG_{0,8}$ | $LG_{0,9}$ | $LG_{0,10}$ | $LG_{0,11}$ | $LG_{0,12}$ | $LG_{0,13}$ | $LG_{0,14}$ |
| Maximum output power (W) | 0.550 | 0.492 | 0.431 | 0.393 | 0.364 | 0.332 | 0.291 |
| Optical−optical Efficiency (%) | 5.50 | 4.92 | 4.31 | 3.93 | 3.64 | 3.32 | 2.91 |

Of note, due to the experimental alignment errors of the elements, including the optical components and the mechanical displacement stage, a slight difference still exists between the results of the experiment and the simulation. In the experiment, the maximal off-axis displacement is limited within 350 μm and the highest $LG_{0,14}$ vortex beam is realized. According to the MORA method, a gain medium with larger size and an off-axis displacement with larger range might help in achieving the high-order mode vortex beam.

## 5. Discussion

The method based on an extra-cavity AMC and IMEs to generate the mode-tunable vortex beam is a typical method [10–15]. In this paper, the proposed MORA method could be used to investigate the theoretical relationship between the modes of the vortex beam and modulation parameters of IMEs for a designed vortex laser. Different from the experimental investigation to generate the mode-tunable vortex beam [10–15], the theoretical relationship

obtained by the MORA method in simulation could be used to help researchers in particularly obtaining a target vortex beam by selecting suitable IMEs and setting suitable modulation parameters of IMEs, which avoids the enormous experimental exploration and waste of time. If a target vortex beam could not be generated from a designed laser with inherent defect, it could be clearly and quickly judged from the obtained relationship rather than the extensive replicated experiments. Moreover, the investigation of the theoretical relationship based on the MORA method could avoid experimental limitations and complex operations, including the modulation values or the accuracy of off-axis displacement [10,13], the modulation ranges of driving voltages [12,14,15], the optimization of structure parameters, etc. The property of the MORA method could help researchers in designing a suitable laser or optimizing a laser. In particular, the influence of the modulation accuracy of IMEs on the generated vortex beams could also be investigated and help researchers in selecting the IMEs with a suitable modulation accuracy to generate the target vortex beam.

As the simulation shown in Section 3.2, the theoretical relationship between the modes of the vortex beam and the off-axis displacement of *OC* is investigated and revealed by the MORA method. It clearly shows that the modes could be generated from the laser within the simulation conditions. Meanwhile, of note, the influence of different increments of *OC* off-axis displacement on the output is investigated and the stable region for a target vortex beam is obtained, which could be used to help researchers in selecting a suitable modulation accuracy of *OC* to obtain a target vortex beam. It is well known that the mode generated from a resonant cavity depends on the intra-cavity gain and loss. For the vortex laser described in the paper, there is a stable region for each generated mode, within which the intra-cavity loss brought by the off-axis displacement of the *OC* could be considered as unchanging. If the off-axis displacement exceeds the stable region, the intra-cavity loss will be distinct and cause the beam mode to change. As the intra-cavity loss is related to the structure of the laser cavity, it is possible to reduce the range and increase the availability of more vortex modes by optimizing the cavity structure (e.g., the curvature radius of the *OC*, length of the resonant cavity, and position of the gain medium).

The efficiency and maximum power of the generated vortex laser beam are determined by the gain and loss in the resonant cavity. For different HG modes, the intra-cavity losses and the corresponding efficiencies are different. From the experiment results, the maximum output power of the generated vortex laser is 1.222 W at the $LG_{0,1}$ mode with 12.22% optical−optical conversion efficiency. If the high power vortex laser is required for certain applications, several methods could be adopted, including optimizing the cavity structure, using the gain medium of larger size, and improving the pumping power.

## 6. Conclusions

In conclusion, a novel MORA method to investigate the relationship between the tunable vortex beam mode and the modulation parameters of a vortex laser is proposed and the validity is verified by simulation and experiment. The presented MORA method consists of three major processes, including the resonant cavity equivalent, intra-cavity iteration and light field output, as well as vortex beam generation and identification. In the MORA analysis method, the resonant cavity equivalent is first carried out to obtain the equivalent straight cavity for a given cavity. The light field, propagating through all of the optical mediums, outputs from the equivalent straight cavity after the intra-cavity propagation iteration process and is converted to a vortex beam by an AMC. Finally, a recognition algorithm is carried out to identify the mode of the vortex beam. In the simulation, a mode-tunable vortex laser is used as the analysis object, consisting of a classic plano-concave straight cavity, a vortex beam generation beamline, and a reference beam acquisition beamline, while the *OC* mirror is set as the intra-cavity modulation element and the off-axis pumping is achieved by adjusting the off-axis displacement of the *OC* mirror. In the simulation, the theoretical relationship between the modes of the vortex beam and modulation parameters is investigated, including the vortex beam generation abilities, the influence of the modulation accuracy, and the stable region of the tuned mode. In the experiment, a mode-tunable vortex laser with primary

parameters, which is set as the same with the simulation model, is built. In the experiment, within the displacement range, the HG beam mode could be continuously tuned from the 1st to the 14th row, and the converted vortex beam from $1\hbar$ to $14\hbar$ could be continuously obtained. The experiment results agree well with the simulation and the validity of the MORA method is verified. Based on the theoretical relationship obtained by the MORA method, it is possible to anticipate the modes of the vortex beam generated from a given cavity based on the modulation of the IMEs. Moreover, the MORA method could be used as a guidance for researchers in optimizing the practical vortex laser to obtain the target vortex beam by the configuration optimization and parameter selection.

**Author Contributions:** Conceptualization, L.H.; data curation, S.L.; formal analysis, S.L.; funding acquisition, L.H.; investigation, S.L.; methodology, S.L.; project administration, L.H.; resources, D.W. and L.H.; software, S.L. and S.K.; supervision, L.H.; validation, D.W. and Y.Z.; visualization, S.L.; writing—original draft, S.L.; writing—review and editing, S.L. and L.H. All authors have read and agreed to the published version of the manuscript.

**Funding:** This research was funded by the National Natural Science Foundation of China, grant number 61775112.

**Institutional Review Board Statement:** Not applicable.

**Informed Consent Statement:** Not applicable.

**Data Availability Statement:** The data presented in this study are available on request from the corresponding author.

**Acknowledgments:** The authors are grateful to the other colleagues for their help during the period of the study.

**Conflicts of Interest:** The authors declare no conflict of interest.

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
