# Peer review of "Theoretical and Experimental Research on the Mode Modulation Regulation for the Mode-Tunable Vortex Laser Based on Mode Conversion and Intra-Cavity Modulation"

_photonics, doi:10.3390/photonics9040232_

Round 1

Reviewer 1 Report

Dear authors,

  1. Some mistakes should  be corrected. The word "filed" in line 158 should be replaced by "field". The parameter "134 um" of dx shown in line 520 can't be found in Fig. 16, Maybe it should be "132 um".
  2.  The title is inappropriate. The mode directed output from the off-axis displacement laser is Hermite-Gaussian mode, rather than the vortex mode. The tunable vortex mode is obtained by converting the Hermite-Gaussian mode using an extra-cavity AMC. The present title will cause trouble to readers. Since the tunable vortex mode can also be directly generated from the hollow spot pumped solid state laser, where different closing degree of the resoantor (or intracavity etalon) and defects on the output coupler can  be used to select the mode. 

Reviewer 2 Report

In this paper, the authors theoretically and experimentally investigate the mode-tunable laser. This work is interesting and worth of publication. In addition, it is well written and organized. However, the following comments should be addressed before publication. 

  1. There are too many abbreviations in the manuscript, which make it difficult to read. Some abbreviations don’t appear often, it is not necessary to use abbreviation, such as, CEE, CE, PE, etc. Moreover, some of them are labeled in everywhere. For example, in the 86th line, mode modulation regulation acquisition (MORA) appears the first time, however, it is repeated in the 133th line as “mode modulation regulation acquisition (MORA)”.
  2. In the 168th line, the horizontal arrows’ color should be pink rather than orange.
  3. The equation number should be (2) not (1) below fig. 3.
  4. In Figs. 12 and 16, we can see that the desired mode can be got in wide range of off-axis displacement ~15 mm, what is the physical reason about this? It should be discussed in the paper. Is it possible to reduce this range and increase the availability of more vortex modes?
  5. If the proposed scheme for generating vortex laser is going to find applications, what is the efficiency and maximum power of the generated vortex laser? And is the efficiency different in different vortex mode? These concerns are important for the integrity of this work.
